# LOGIC-BASED ADAPTIVE REWARD SHAPING FOR REINFORCEMENT LEARNING

## ABSTRACT

There is a growing interest in using formal languages such as Linear Temporal Logic (LTL) to specify complex tasks and reward functions for reinforcement learning (RL) precisely and succinctly. Nevertheless, existing methods often assign sparse rewards, which may require millions of exploratory episodes to converge to a quality policy. To address this limitation, we adopt the notion of *task progression* to measure the degree to which a task specified by a co-safe LTL formula is partially completed and design several reward functions to incentivize a RL agent to satisfy the task specification as much as possible. We also develop an adaptive reward shaping approach that dynamically updates reward functions during the learning process. Experimental results on a range of benchmark RL environments demonstrate that the proposed approach generally outperforms baselines, achieving earlier convergence to a policy with a higher success rate of task completion and a higher normalized expected discounted return.

## 1 INTRODUCTION

In reinforcement learning (RL), an agent's behavior is guided by reward functions, which are often difficult to specify manually when representing complex tasks. Alternatively, a RL agent can infer the intended reward from demonstrations (Ng & Russell, 2000), trajectory comparisons (Wirth et al., 2017), or human instructions (Fu et al., 2018). Recent years have witnessed a growing interest in using formal languages such as Linear Temporal Logic (LTL) and automata to specify complex tasks and reward functions for RL precisely and succinctly (Li et al., 2017; Icarte et al., 2018; Camacho et al., 2019; De Giacomo et al., 2019; Jothimurugan et al., 2019; Bozkurt et al., 2020; Hasanbeig et al., 2020; Jiang et al., 2021; Icarte et al., 2022; Cai et al., 2023).

Nevertheless, existing methods often assign rewards sparsely (e.g., an agent receives reward 1 only if a task specification is satisfied or an accepting automaton state has been reached, and receives reward 0 otherwise). Sparse rewards may necessitate millions of exploratory episodes for convergence to a quality policy. Moreover, some prior works are only compatible with specific RL algorithms tailored for their proposed reward structures, such as *q-learning for reward machines* (QRM) described in (Camacho et al., 2019), *counterfactual experiences for reward machines* (CRM) developed in (Icarte et al., 2022), and the modular DDPG proposed in (Hasanbeig et al., 2020).

*Reward shaping* (Ng et al., 1999) is a paradigm where an agent receives some intermediate rewards as it gets closer to the goal and has shown to be helpful for RL algorithms to converge more quickly. Inspired by this idea, we develop a logic-based adaptive reward shaping approach in this work. We use the syntactically co-safe fragment of LTL to specify complex RL tasks, such as "the agent should collect an orange flag and a blue flag in any order while avoiding the yellow flag" and "the agent should get coffee and mail before delivering them to the office". We then translate a co-safe LTL task specification into a deterministic finite automaton (DFA) and build a product of the DFA and the environment model represented as a *Markov decision process* (MDP). We design reward functions for the resulting product MDP that keeps track of the task completion status (e.g., a task is completed if an accepting state of the DFA has been reached).

The principle underlying our approach is that we want to assign intermediate rewards to an agent as it makes progress toward completing a task. A key challenge is how to measure the closeness to task completion. To this end, we adopt the notion of *task progression* defined by Lacerda et al. (2019),

which measures each DFA state's distance to acceptance. The smaller the distance, the higher degree of task progression. The distance is zero when the task is fully completed.

Another challenge is what reward values to assign for various degrees of task progression. We design three different reward functions. The *naive* reward function assigns reward 1 to each transition that reduces the distance to acceptance. The *progression* reward function assigns reward values based on the degree to which the distance to acceptance has been reduced in a transition. The *hybrid* reward function balances the progression reward and the penalty for self-loops (i.e., staying in the same DFA state). However, these reward functions may yield optimal policies where a task is only partially completed.

To address this issue, we develop an adaptive reward shaping approach that dynamically updates distance to acceptance values during the learning process, accounting for information obtained from DFA executions in recent training episodes. We design two new reward functions, namely *adaptive progression* and *adaptive hybrid*, which can leverage the updated distance to acceptance values. We demonstrate via examples that an optimal policy satisfying the task specification can be learnt within several rounds of updates.

We evaluate the proposed approach on a range of benchmark RL environments and compare with two baselines (i.e., QRM (Camacho et al., 2019) and CRM (Icarte et al., 2022)). To demonstrate that our approach is agnostic to RL algorithms, we use DQN (Mnih et al., 2015), DDQN (Van Hasselt et al., 2016), DDPG (Lillicrap et al., 2016), A2C (Mnih et al., 2016), and PPO (Schulman et al., 2017) in the experiments. Results show that the proposed approach generally outperforms baselines, achieving earlier convergence to a policy with a higher success rate of task completion and a higher normalized expected discounted return.

## 2 RELATED WORK

Li et al. (2017) presented one of the first works applying temporal logic to reward function design. A variant of temporal logic called truncated LTL was proposed for specifying tasks. Reward functions are obtained via checking robustness degrees of satisfying truncated LTL formulas. This method is limited to Markovian rewards, while our approach can generate non-Markovian rewards.

There is a line of work on *reward machines (RM)* (Icarte et al., 2018), which is a type of finite state machine that takes labels representing environment abstractions as input, and outputs reward functions. Camacho et al. (2019) showed that LTL and other regular languages can be automatically translated into RMs via the construction of DFAs; they also presented a technique named QRM with reward shaping, which tailors q-learning for RMs and computes a potential function over RM states for reward shaping. Icarte et al. (2022) developed another tailored method named CRM, which generates synthetic experiences for learning via counterfactual reasoning. We adopt QRM and CRM (with reward shaping) as baselines for comparison in our experiments. As we will show in Section 5, the performance of these methods suffers from reward sparsity.

De Giacomo et al. (2019) used a fragment of LTL for finite traces (called $LTL_f$) to encode RL rewards. There are also several methods seeking to learn optimal policies that maximize the probability of satisfying a LTL formula (Hasanbeig et al., 2019; Bozkurt et al., 2020; Hasanbeig et al., 2020). However, none of these methods assigns intermediate rewards for task progression.

Jothimurugan et al. (2019) proposed a new specification language which can be translated into reward functions. Their method uses a task monitor to track the degree of specification satisfaction and assign intermediate rewards. But they require users to encode atomic predicates into quantitative values for reward assignment. By contrast, our approach automatically assigns intermediate rewards using DFA states' distance to acceptance values, eliminating the need for user-provided functions.

Jiang et al. (2021) presented a reward shaping framework for average-reward learning in continuing tasks. Their method automatically translates a LTL formula encoding domain knowledge into a function that provides additional reward throughout the learning process. This work has a different problem setup and thus is not directly comparable with our approach.

Cai et al. (2023) proposed a model-free RL method for minimally-violating an infeasible LTL specification. Their method also considers the assignment of intermediate rewards, but their definition of task progression requires additional information about the environment (e.g., geometric distance

from each waypoint to the destination). By contrast, we define task progression based on the task specification only, following (Lacerda et al., 2019) which is a work on robotic planning with MDPs (but not RL).

## 3 BACKGROUND

### 3.1 REINFORCEMENT LEARNING

We consider a RL agent that interacts with an unknown environment modeled as an episodic MDP where each learning episode terminates within a finite horizon $H$. Formally, an MDP is denoted as a tuple $\mathcal{M} = (S, s_0, A, T, R, \gamma, AP, L)$ where $S$ is a finite set of states, $s_0 \in S$ is an initial state, $A$ is a finite set of actions, $T : S \times A \times S \rightarrow [0, 1]$ is a probabilistic transition function, $R$ is a reward function, $\gamma \in [0, 1]$ is a discount factor, $AP$ is a finite set of atomic propositions, and $L : S \rightarrow 2^{AP}$ is a labeling function. At each time $t$ of an episode, the agent selects an action $a_t$ in state $s_t$ following a policy $\pi$, ends in state $s_{t+1}$ drawn from the probability distribution $T(\cdot|s_t, a_t)$ and receives a reward $r_t$. The agent seeks to learn an optimal policy $\pi^*$ that maximizes the expected discounted return $\mathbb{E}_\pi[\sum_{i=0}^{H-t} \gamma^i r_{t+i}|s_t = s]$ when starting from any state $s \in S$ at time $t$.

The reward function can be Markovian, denoted by $R : S \times A \times S \rightarrow \mathbb{R}$, or non-Markovian (i.e., history dependent), denoted by $R : (S \times A)^* \rightarrow \mathbb{R}$. The reward function is unknown to the agent, but it can be specified by the designer to achieve desired agent behavior. In this work, we specify complex RL tasks and design reward functions using temporal logic described below.

### 3.2 TASK SPECIFICATIONS

**Co-safe LTL.** LTL (Pnueli, 1981) is a modal logic that extends the propositional logic with temporal operators. In this work, we use the syntactically co-safe fragment of LTL with the following syntax:

$$\varphi := \alpha \mid \neg\alpha \mid \varphi_1 \wedge \varphi_2 \mid \varphi_1 \vee \varphi_2 \mid \bigcirc\varphi \mid \varphi_1 \mathsf{U} \varphi_2 \mid \Diamond\varphi$$

where $\alpha \in AP$ is an atomic proposition, $\neg$ (negation), $\wedge$ (conjunction), and $\vee$ (disjunction) are Boolean operators, while $\bigcirc$ (next), $\mathsf{U}$ (until), and $\Diamond$ (eventually) are temporal operators. Intuitively, $\bigcirc\varphi$ means that $\varphi$ has to hold in the next step; $\varphi_1 \mathsf{U} \varphi_2$ means that $\varphi_1$ has to hold at least until $\varphi_2$ becomes true; and $\Diamond\varphi$ means that $\varphi$ becomes true at some time eventually.

We can convert a co-safe LTL formula $\varphi$ into a DFA $\mathcal{A}_\varphi$ that accepts exactly the set of good prefixes for $\varphi$, that is, the set of finite sequences satisfying $\varphi$ regardless of how they are completed with any suffix (Kupferman & Vardi, 2001). Let $\mathcal{A}_\varphi = (Q, q_0, Q_F, 2^{AP}, \delta)$ denote the converted DFA, where $Q$ is a finite set of states, $q_0$ is the initial state, $Q_F \subseteq Q$ is a set of accepting states, $2^{AP}$ is the alphabet, and $\delta : Q \times 2^{AP} \rightarrow Q$ is the transition function.

*Example 1.* Consider an agent navigating in a grid world envrionment shown in Figure 1a. The agent's task is to collect an *orange* flag and a *blue* flag (in any order) while avoiding the *yellow* flag. A learning episode ends when the agent completes the task, hits the yellow flag, or reaches 25 steps. We can specify this task with a co-safe LTL formula $\varphi = (\neg y)\mathsf{U}((o\wedge((\neg y)\mathsf{U}b))\vee(b\wedge((\neg y)\mathsf{U}o)))$, where $o$, $b$ and $y$ represent collecting *orange*, *blue* and *yellow* flags, respectively. Figure 1b shows the corresponding DFA $\mathcal{A}_\varphi$, which has five states including the initial state $q_0$ (depicted with an incoming arrow) and the accepting state $Q_F = \{q_4\}$ (depicted with double circle). A transition is enabled when its labelled Boolean formula holds. For instance, the transition $q_0 \rightarrow q_4$ is enabled when both the orange and blue flags have been collected (i.e., $b\wedge o$ is true). Starting from the initial state $q_0$, a path ending in the accepting state $q_4$ represents a good prefix of satisfying $\varphi$ (i.e., the task is completed); and a path ending in the trap state $q_3$ represents that $\varphi$ is violated (i.e., the yellow flag is reached before the task completion). ∎

**Task progression.** We adopt the notion of *task progression* defined by Lacerda et al. (2019) to measure the degree to which a co-safe LTL formula is partially satisfied. Intuitively, we would want to encourage the agent to complete as much of the task as possible.

Given a DFA $\mathcal{A}_\varphi = (Q, q_0, Q_F, 2^{AP}, \delta)$ for a co-safe LTL formula $\varphi$, let $\mathrm{Suc}_q \subseteq Q$ denote the set of successors of state $q$ and let $|\delta_{q,q'}| \in \{0, \ldots, 2^{|AP|}\}$ denote the number of possible transitions

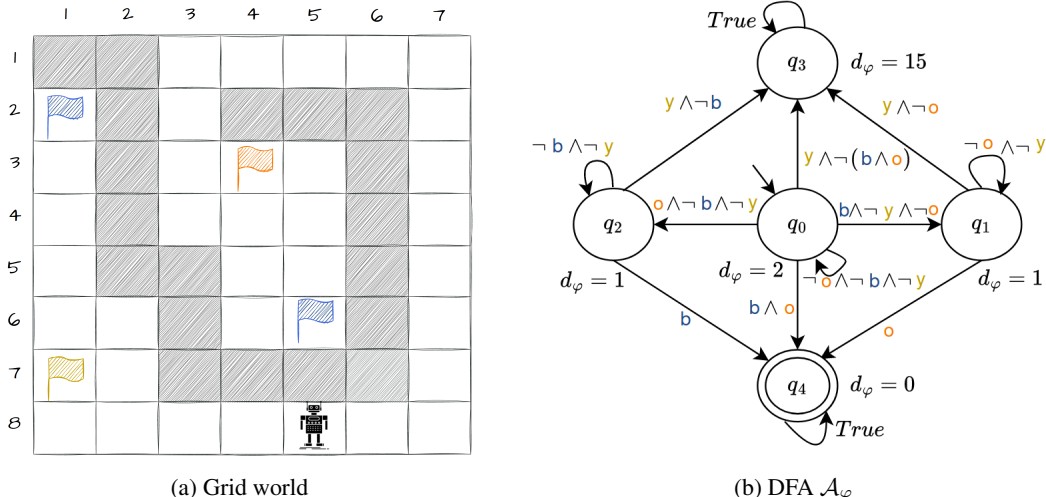

(a) Grid world  (b) DFA $\mathcal{A}_\varphi$

Figure 1: An example grid world environment and a DFA corresponding to a co-safe LTL formula $\varphi = (\neg y)\mathsf{U}((o\wedge((\neg y)\mathsf{U}b))\vee(b\wedge((\neg y)\mathsf{U}o)))$.

from $q$ to $q'$. A *distance to acceptance function* $d_\varphi : Q \to \mathbb{R}_{\geq 0}$ is defined as:

$$
d_\varphi(q) = \begin{cases} 0 & \text{if } q \in Q_F \\ \min_{q' \in \text{Suc}_q} d_\varphi(q') + h(q, q') & \text{if } q \notin Q_F \text{ and } Q_F \text{ is reachable from } q \\ |AP| \cdot |Q| & \text{otherwise} \end{cases} \tag{1}
$$

where $h(q, q') := \log_2\left(\left\{\frac{2^{|AP|}}{|\delta_{q,q'}|}\right\}\right)$ represents the difficulty of moving from $q$ to $q'$ in the DFA $\mathcal{A}_\varphi$. The *progression function* $\rho_\varphi : Q \times Q \to \mathbb{R}_{\geq 0}$ is then defined as follows:

$$
\rho_\varphi(q, q') = \begin{cases} \max\{0, d_\varphi(q) - d_\varphi(q')\} & \text{if } q' \in \text{Suc}_q \text{ and } q' \not\to^* q \\ 0 & \text{otherwise} \end{cases} \tag{2}
$$

where $q' \not\to^* q$ represents that there does not exist a path from $q'$ to $q$ in the DFA $\mathcal{A}_\varphi$ (to avoid cycles in $\mathcal{A}_\varphi$ with non-zero progression values).

*Example 2.* Each state in the DFA $\mathcal{A}_\varphi$ shown in Figure 1b is annotated with its distance to acceptance function value. We have $d_\varphi(q_4) = 0$ because $q_4$ is an accepting state, and $d_\varphi(q_3) = 3 \times 5 = 15$ because $q_3$ is a trap state. We compute the distance to acceptance values for the rest of the states recursively following Equation 1, and obtain $d_\varphi(q_0) = 2$, $d_\varphi(q_1) = 1$, and $d_\varphi(q_2) = 1$. The progression value $\rho_\varphi(q_0, q_1) = \max\{0, d_\varphi(q_0) - d_\varphi(q_1)\} = 1$, indicating that a positive progression has been made toward the task completion; while $\rho_\varphi(q_0, q_3) = \max\{0, d_\varphi(q_0) - d_\varphi(q_3)\} = 0$, since moving to the trap state $q_3$ does not result in any task progression. $\blacksquare$

**MDP-DFA product.** Given an MDP $\mathcal{M} = (S, s_0, A, T, R, \gamma, AP, L)$ and a DFA $\mathcal{A}_\varphi = (Q, q_0, Q_F, 2^{AP}, \delta)$ corresponding to a co-safe LTL formula $\varphi$, we define their product MDP as $\mathcal{M}^\otimes = \mathcal{M} \otimes \mathcal{A}_\varphi = (S^\otimes, s_0^\otimes, A, T^\otimes, R^\otimes, \gamma, AP, L^\otimes)$ where $S^\otimes = S \times Q$, $s_0^\otimes = \langle s_0, \delta(q_0, L(s_0)) \rangle$,

$$
T^\otimes(\langle s, q \rangle, a, \langle s', q' \rangle) = \begin{cases} T(s, a, s') & \text{if } q' = \delta(q, L(s)) \\ 0 & \text{otherwise} \end{cases}
$$

and $L^\otimes(\langle s, q \rangle) = L(s)$.

We design Markovian reward functions $R^\otimes : S^\otimes \times A \times S^\otimes \to \mathbb{R}$ for the product MDP, whose projection onto the MDP $\mathcal{M}$ yields non-Markovian reward functions. The projected reward function $R$ is Markovian only if $|Q| = 1$ (i.e., the DFA has one state only).

## 4 APPROACH

We present a range of reward functions to incentivize the RL agent to complete a task specified by a co-safe LTL formula as much as possible (cf. Section 4.1), and develop an adaptive reward shaping approach that dynamically updates the reward functions during the learning process (cf. Section 4.2).

### 4.1 REWARD FUNCTIONS FOR PARTIALLY SATISFIABLE TASK SPECIFICATIONS

**Naive reward function.** First, we consider a naive way of rewarding each transition that reduces the distance to acceptance. We define a *naive reward function* for the product MDP $\mathcal{M}^{\otimes} = \mathcal{M} \otimes \mathcal{A}_{\varphi}$:

$$R_{\mathsf{nv}}^{\otimes}\left(\langle s, q\rangle, a, \langle s', q'\rangle\right) = \begin{cases} 1 & \text{if } d_{\varphi}(q) > d_{\varphi}(q') \text{ and } Q_F \text{ is reachable from } q' \\ 0 & \text{otherwise} \end{cases} \quad (3)$$

*Example 3.* Consider the following candidate policies for the RL agent navigating in the grid world environment shown in Figure 1a. Let $g_{ij}$ denote the grid in row $i$ and column $j$. Suppose the agent's initial location is $g_{85}$.

- $\pi_1$: The agent moves 10 steps to collect the blue flag in $g_{21}$ while avoiding the yellow flag in $g_{71}$. But it fails to reach the orange flag in $g_{34}$ before the episode time-out (25 steps).
- $\pi_2$: The agent moves 16 steps to collect the orange flag in $g_{34}$ and then moves 4 more steps to collect the blue flag in $g_{65}$. The task is completed.
- $\pi_3$: The agent moves directly to the yellow flag in 5 steps. The episode ends.

Suppose the grid world is deterministic (i.e., $T(s, a, s')$ is a Dirac distribution) and the discount factor is $\gamma = 0.9$. The initial state of the product MDP is $s_0^{\otimes} = \langle g_{85}, q_0 \rangle$. With the naive reward function, we have $V_{\mathsf{nv}}^{\pi_1}(s_0^{\otimes}) = 0.9^9 \approx 0.39$, $V_{\mathsf{nv}}^{\pi_2}(s_0^{\otimes}) = 0.9^{15} + 0.9^{19} \approx 0.34$, and $V_{\mathsf{nv}}^{\pi_3}(s_0^{\otimes}) = 0$. Thus, the agent would choose $\pi_1$ as the optimal policy because of the maximal expected discounted return, but it only satisfies the task partially. ∎

**Progression reward function.** Next, we define a *progression reward function* based on the task progression function introduced in Equation 2, to account for the degree to which the distance to acceptance has been reduced.

$$R_{\mathsf{pg}}^{\otimes}\left(\langle s, q\rangle, a, \langle s', q'\rangle\right) = \rho_{\varphi}(q, q') = \begin{cases} \max\{0, d_{\varphi}(q) - d_{\varphi}(q')\} & \text{if } q' \in \mathrm{Suc}_q \text{ and } q' \not\rightarrow^* q \\ 0 & \text{otherwise} \end{cases} \quad (4)$$

*Example 4.* We evaluate the three policies described in Example 3 with the progression reward function. We have $V_{\mathsf{pg}}^{\pi_1}(s_0^{\otimes}) = 0.9^9 \approx 0.39$, $V_{\mathsf{pg}}^{\pi_2}(s_0^{\otimes}) = 0.9^{15} + 0.9^{19} \approx 0.34$, and $V_{\mathsf{pg}}^{\pi_3}(s_0^{\otimes}) = 0$. Thus, the agent would choose the optimal policy $\pi_1$. ∎

**Hybrid reward function.** So far, we have only considered rewarding transitions that progress toward acceptance and do not penalize transitions that remain in the same DFA state. To address this issue, we define a *hybrid reward function* for the product MDP $\mathcal{M}^{\otimes} = \mathcal{M} \otimes \mathcal{A}_{\varphi}$ as follows:

$$R_{\mathsf{hd}}^{\otimes}\left(\langle s, q\rangle, a, \langle s', q'\rangle\right) = \begin{cases} \eta \cdot -d_{\varphi}(q) & \text{if } q = q' \\ (1 - \eta) \cdot \rho_{\varphi}(q, q') & \text{otherwise} \end{cases} \quad (5)$$

where $\eta \in [0, 1]$ is a parameter to balance the trade-offs between penalties and progression rewards.

*Example 5.* We evaluate policies in Example 3 with the hybrid reward function (suppose $\eta = 0.1$). We have $V_{\mathsf{hd}}^{\pi_1}(s_0^{\otimes}) = -2\eta \cdot \sum_{i=0}^{8} \gamma^i + (1 - \eta) \cdot \gamma^9 - \eta \cdot \sum_{i=10}^{24} \gamma^i \approx -1.15$, $V_{\mathsf{hd}}^{\pi_2}(s_0^{\otimes}) = -2\eta \cdot \sum_{i=0}^{14} \gamma^i + (1-\eta) \cdot \gamma^{15} - \eta \cdot \sum_{i=16}^{18} \gamma^i + (1-\eta) \cdot \gamma^{19} \approx -1.33$, and $V_{\mathsf{hd}}^{\pi_3}(s_0^{\otimes}) = -2\eta \cdot \sum_{i=0}^{3} \gamma^i \approx -0.69$. Thus, the agent would choose $\pi_3$ as the optimal policy, under which it bumps into the yellow flag directly to avoid further penalties. Increasing the value of $\eta$ would weight more on penalties and not change the optimal policy in this example, while decreasing the value of $\eta$ tends to the progression reward function (special case with $\eta = 0$). ∎

## 4.2 ADAPTIVE REWARD SHAPING

Although reward functions proposed in Section 4.1 incentivize the RL agent to complete a task specified with a co-safe LTL formula as much as possible, Examples 3, 4 and 5 show that optimal policies chosen by the agent may not satisfy the co-safe LTL specification. One possible reason is that the distance to acceptance function $d_\varphi$ (cf. Equation 1) may not accurately represent the difficulty to trigger the desired DFA transitions in a given environment. To tackle this limitation, we develop an adaptive reward shaping approach that dynamically updates distance to acceptance values and reward functions during the learning process.

**Updating the distance to acceptance values.** After every $N$ learning episodes where $N$ is a domain-specific parameter, we check the average success rate of task completion (i.e., an episode is successful if it ends in an accepting state of the DFA $\mathcal{A}_\varphi$). If the average success rate drops below a certain threshold $\lambda$, we update the distance to acceptance values. We obtain the initial values $d_\varphi^0(q)$ for each DFA state $q \in Q$ based on Equation 1. We compute the distance to acceptance values for the $k$-th round of updates recursively as follows:

$$d_\varphi^k(q) = \begin{cases} \mu \cdot d_\varphi^{k-1}(q) + \theta & \text{if } q \text{ is a trap state} \\ \mu \cdot d_\varphi^{k-1}(q) + \theta \cdot Pr(q|\sigma) & \text{otherwise} \end{cases} \tag{6}$$

where $\mu \in [0,1]$ is a weight value indicating the extent to which the current distance values should be preserved, $Pr(q|\sigma)$ is the probability of state $q$ occurring in the DFA executions $\sigma$ during the past $N$ learning episodes (which can be obtained via frequency counting), and $\theta \geq \sum_{q \in Q} d_\varphi^0(q)$ is a domain-specific parameter. Intuitively, the agent would experience more difficulty to reach accepting states if it is stuck in state $q$ more often as indicated by a greater value of $Pr(q|\sigma)$.

*Example 6.* Suppose $N = 1$, $\mu = 0.5$ and $\theta = 25$. The initial distance to acceptance values are $d_\varphi^0(q_0) = 2$, $d_\varphi^0(q_1) = d_\varphi^0(q_2) = 1$, $d_\varphi^0(q_3) = 15$, and $d_\varphi^0(q_4) = 0$ following Example 2. Suppose the agent's movement during the first episode follows policy $\pi_1$. We update the distance to acceptance values as $d_\varphi^1(q_0) = 0.5 \times 2 + 25 \times \frac{9}{25} = 10$, $d_\varphi^1(q_1) = 0.5 \times 1 + 25 \times \frac{16}{25} = 16.5$, $d_\varphi^1(q_2) = 0.5 \times 1 + 0 = 0.5$, $d_\varphi^1(q_3) = 0.5 \times 15 + 25 = 32.5$, and $d_\varphi^1(q_4) = 0.5 \times 0 + 0 = 0$. ∎

The ordering of DFA states based on the updated distance to acceptance values may change from round to round. In the above example, $d_\varphi^0(q_0) > d_\varphi^0(q_1)$, but $d_\varphi^1(q_0) < d_\varphi^1(q_1)$. Thus, we cannot use reward functions proposed in Section 4.1 directly for adaptive reward shaping. To this end, we present two new reward functions as follows.

**Adaptive progression reward function.** Given the updated distance to acceptance values $d_\varphi^k(q)$ for each DFA state $q \in Q$, we apply the progression function defined in Equation 2 and obtain

$$\rho_\varphi^k(q, q') = \begin{cases} \max\{0, d_\varphi^k(q) - d_\varphi^k(q')\} & \text{if } q' \in \text{Suc}_q \text{ and } q' \not\to^* q \\ 0 & \text{otherwise} \end{cases} \tag{7}$$

Then, we define an *adaptive progression reward function* for the $k$-th round of updates as:

$$R_{\text{ap},k}^\otimes(\langle s, q \rangle, a, \langle s', q' \rangle) = \max\{\rho_\varphi^0(q, q'), \rho_\varphi^k(q, q')\} \tag{8}$$

When $k = 0$, the adaptive progression reward function $R_{\text{ap},0}^\otimes$ coincides with the progression reward function $R_{\text{pg}}^\otimes$ defined in Equation 4.

*Example 7.* We compute $R_{\text{ap},1}^\otimes$ for $k = 1$ using distance to acceptance values obtained in Example 6. For instance, $R_{\text{ap},1}^\otimes(\langle g_{31}, q_0 \rangle, north, \langle g_{21}, q_1 \rangle) = \max\{\rho_\varphi^0(q_0, q_1), \rho_\varphi^1(q_0, q_1)\} = \max\{1, 0\} = 1$ and $R_{\text{ap},1}^\otimes(\langle g_{33}, q_0 \rangle, east, \langle g_{34}, q_2 \rangle) = \max\{\rho_\varphi^0(q_0, q_2), \rho_\varphi^1(q_0, q_2)\} = \max\{1, 9.5\} = 9.5$. We evaluate policies described in Example 3 with the adaptive progression reward function. We have $V_{\text{ap},1}^{\pi_1}(s_0^\otimes) = 0.9^9 \approx 0.39$, $V_{\text{ap},1}^{\pi_2}(s_0^\otimes) = 9.5 \times 0.9^{15} + 1 \times 0.9^{19} \approx 2.09$, and $V_{\text{ap},1}^{\pi_3}(s_0^\otimes) = 0$. Thus, the agent would choose the optimal policy $\pi_2$ which not only maximizes the expected discounted return but also completes the task specified by the co-safe LTL formula $\varphi$. ∎

**Adaptive hybrid reward function.** We define an *adaptive hybrid reward function* for the $k$-th round of updates as:

$$R_{\mathsf{ah},k}^{\otimes}\left(\langle s, q\rangle, a, \langle s', q'\rangle\right) = \begin{cases} \eta_k \cdot -d_{\varphi}^k(q) & \text{if } q = q' \\ (1 - \eta_k) \cdot \max\{\rho_{\varphi}^0(q, q'), \rho_{\varphi}^k(q, q')\} & \text{otherwise} \end{cases} \quad (9)$$

where $\eta_0 \in [0, 1]$ and $\eta_k = \frac{\eta_{k-1}}{2\theta}$. We adjust the weight value $\eta_k$ in each round of updates to prevent undesired agent behavior due to the increased penalty for self-loops. When $k = 0$, the adaptive hybrid reward function $R_{\mathsf{ah},0}^{\otimes}$ coincides with the hybrid reward function $R_{\mathsf{hd}}^{\otimes}$ defined in Equation 5.

*Example 8.* Suppose $N = 1$, $\mu = 0.5$, $\theta = 25$, and $\eta_0 = 0.1$. The initial distance to acceptance values $d_{\varphi}^0$ are the same as in Example 6. Suppose the agent's movement during the first episode follows policy $\pi_3$, which is the optimal policy obtained using the hybrid reward function in Example 5. We update the distance to acceptance values as $d_{\varphi}^1(q_0) = 0.5 \times 2 + 25 \times \frac{4}{5} = 21$, $d_{\varphi}^1(q_1) = d_{\varphi}^1(q_2) = 0.5$, $d_{\varphi}^1(q_3) = 32.5$, and $d_{\varphi}^1(q_4) = 0$. We compute $R_{\mathsf{ah},1}^{\otimes}$ with $\eta_1 = 0.002$, which yields $V_{\mathsf{ah},1}^{\pi_1}(s_0^{\otimes}) \approx 7.41$, $V_{\mathsf{ah},1}^{\pi_2}(s_0^{\otimes}) \approx 3.68$, and $V_{\mathsf{ah},1}^{\pi_3}(s_0^{\otimes}) \approx -0.28$. We continue the process of learning with the adaptive reward shaping until the 4th round of updates where we obtain $d_{\varphi}^4(q_0) = 28.88, d_{\varphi}^4(q_1) = 14.65, d_{\varphi}^4(q_2) = 0.06, d_{\varphi}^4(q_3) = 47.81, d_{\varphi}^4(q_4) = 0$ and $\eta_4$ close to 0. We compute $R_{\mathsf{ah},4}^{\otimes}$ and obtain $V_{\mathsf{ah},4}^{\pi_1}(s_0^{\otimes}) \approx 5.51$, $V_{\mathsf{ah},4}^{\pi_2}(s_0^{\otimes}) \approx 6.07$, and $V_{\mathsf{ah},4}^{\pi_3}(s_0^{\otimes}) \approx 0$, which lead to the optimal policy $\pi_2$ being chosen by the agent. ∎

In summary, an optimal policy that maximizes the expected discounted return does not necessarily satisfy the task specification (cf. Examples 3, 4 and 5). Examples 7 and 8 show that an optimal policy under which the task is completed could be learnt within several updates of adaptive progression and adaptive hybrid reward functions, respectively.

## 5 EXPERIMENTS

**RL domains.** We empirically evaluate the proposed approach with the following benchmark RL domains. The taxi domain is taken from OpenAI gym (Brockman et al., 2016), while the other three domains are adapted from (Icarte et al., 2022).

- *Office world*: The agent navigates in a $12 \times 9$ grid world to complete the following task: get coffee and mail (in any order) and deliver them to the office while avoiding obstacles. The test environment assigns reward 1 for each subgoal: (i) get coffee, (ii) get coffee and mail, and (iii) deliver coffee and mail to office, during all of which obstacles should be avoided.

- *Taxi world*: The agent drives around a $5 \times 5$ grid world to pickup and drop off a passenger. The agent starts off at a random location. There are five possible pickup locations and four possible destinations. The task is completed when the passenger is dropped off at the target destination. The test environment assigns reward 1 for each subgoal: (i) pickup a passenger, (ii) reach the target destination with the passenger, and (iii) drop off the passenger.

- *Water world*: The agent moves in a continuous two-dimensional box with floating balls in six colors. The agent's velocity toward one of the four cardinal directions may change at each step. The task is to touch red and green in strict order without touching other colored balls, followed by touching blue balls. The test environment assigns reward 1 for touching each target ball.

- *HalfCheetah*: The agent is a cheetah-like robot with a continuous action space and learns to control six joints to move forward or backward. The task is completed when the agent reaches the farthest location. The test environment assigns reward 1 for reaching each of the five locations along the way.

For each domain, we consider three types of environments: (1) *deterministic* environments, where MDP transitions follow Dirac distributions; (2) *noisy* environments, where each action has certain control noise; and (3) *infeasible* environments, where some subgoals are infeasible to complete (e.g., a blocked office not accessible by the agent, or missing blue balls in the water world).

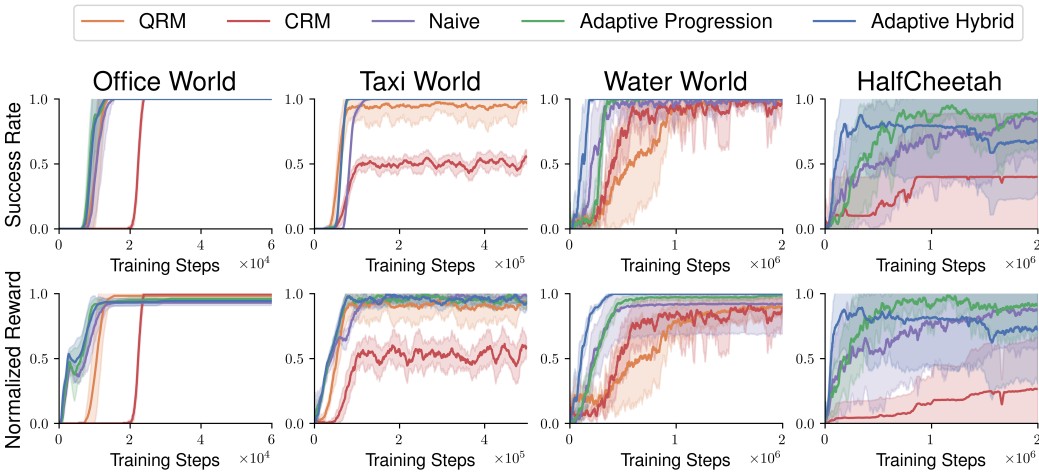

Figure 2: Results in deterministic environments.

**Baselines and metrics.** We compare the proposed approach with two baseline methods: *Q-learning for reward machines* (QRM) with reward shaping (Camacho et al., 2019) and *counterfactual experiences for reward machines* (CRM) with reward shaping (Icarte et al., 2022). We use the code accompanying publications.

We use DQN (Mnih et al., 2015) for learning in discrete domains (office world and taxi world), DDQN (Van Hasselt et al., 2016) for water world with continuous state space, and DDPG (Lillicrap et al., 2016) for HalfCheetah with continuous action space. Note that QRM does not work with DDPG, so we only use CRM as the baseline for HalfCheetah. We also apply A2C (Mnih et al., 2016) and PPO (Schulman et al., 2017) to HalfCheetah (none of the baselines is compatible with these RL algorithms) and report results in the appendix due to the page limit.

We evaluate the performance with two metrics: *success rate of task completion* and *normalized expected discounted return*. We pause the learning process for every 100 training steps in the office world and every 1,000 training steps in other domains, and evaluate the current policy in the test environment over 5 episodes. We compute the success rate via frequency counting of successful episodes where the task is completed, and calculate the average expected discounted return which is normalized using the maximum discounted return possible on that task. The only exception is taxi world where the maximum discounted return varies for different initial states. We pick a normalization factor by averaging the maximum discounted return of all possible initial states. We report the performance of 10 independent trials for each method.

**Results.** Figures 2, 3 and 4 plot the mean performance with 95% confidence interval (the shadowed area) in deterministic, noisy, and infeasible environments, respectively. We omit to plot the success rate in Figure 4 because they are all zero (i.e., the task is infeasible to complete). Hyperparameters used in experiments are included in the appendix.

We find that the proposed approach using naive, adaptive progression, or adaptive hybrid reward functions generally outperform baselines, achieving earlier convergence to policies with a higher success rate of task completion and a higher normalized expected discounted return. The only outlier is the noisy office world where QRM and CRM outperform our approach. One possible reason is that our approach gets stuck with a suboptimal policy, which goes for fetching coffee in a closer location due to the uncertainty introduced by noisy actions.

The significant advantage of our approach is best illustrated in Figure 4. The baselines are not very effective for learning in infeasible environments, because they assign sparse rewards (e.g. the agent only receives a large reward when it completes the task, which is infeasible due to the environment constraints). By contrast, our approach rewards the agent for partially completing a task. Thus, the agent can still learn to complete a task as much as possible.

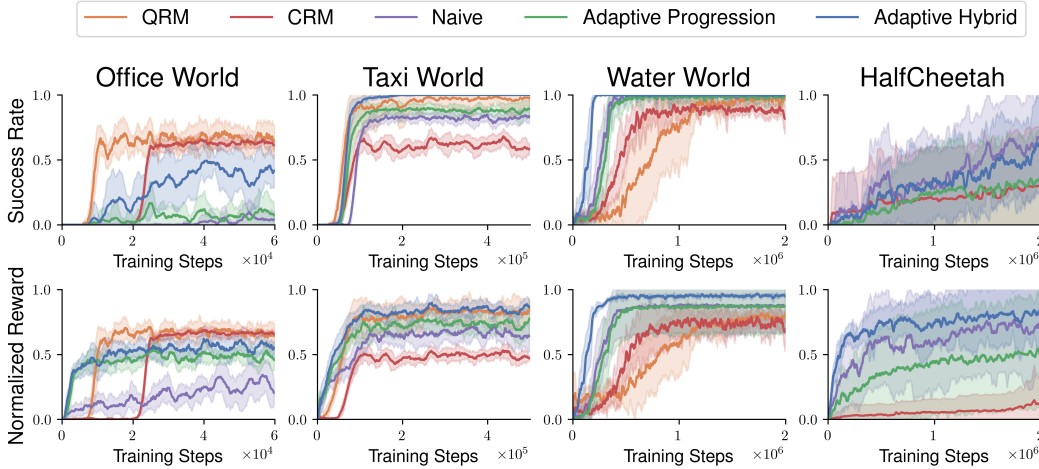

Figure 3: Results in noisy environments

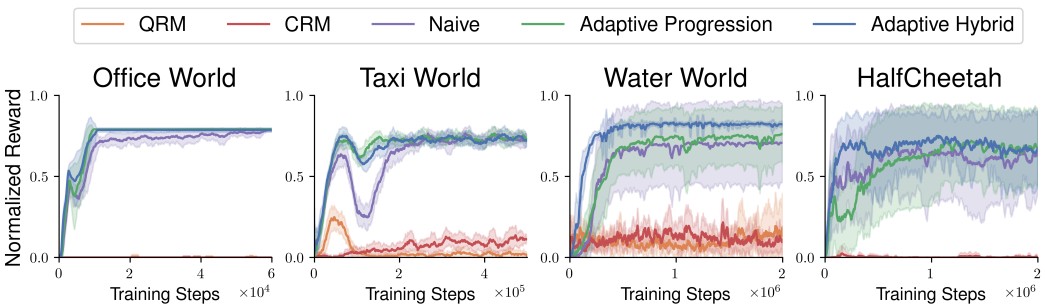

Figure 4: Results in infeasible environments

Comparing among the proposed reward functions, we observe that adaptive hybrid reward function has the best performance in general. Adaptive progression reward function outperforms naive reward function in most environments, while having comparable performance in others.

# 6 CONCLUSION

We have developed a logic-based adaptive reward shaping approach for RL. Our approach utilizes reward functions that are designed to incentivize an agent to complete a task specified by a co-safe LTL formula as much as possible, and dynamically updates these reward functions during the learning process. We also demonstrate via experiments that our approach is compatible with many different RL algorithms, and can outperform state-of-the-art baselines.

There are several directions to explore for possible future work. First, we will evaluate the proposed approach on a wide range of RL domains, beyond those benchmarks considered in the experiments. Second, we will explore an extension to multi-agent RL. Finally, we would like to apply the proposed approach to RL tasks in real-world scenarios (e.g., autonomous driving).

## REPRODUCIBILITY STATEMENT

To ensure the reproducibility of our work, we conduct evaluation experiments on benchmark RL domains and describe the experimental details in Section 5 and the appendix. We will make the code for our work publicly available in GitHub after the double-blind review process. We plan on providing a link to an anonymous repository by posting a comment directed to the reviewers and area chairs once the discussion forum is open, as suggested by the ICLR 2024 author guide.

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

## A    APPENDIX

We show additional experimental results of the HalfCheetah domain in Section A.1 and describe hyperparameters used in experiments in Section A.2.

### A.1    EXPERIMENTAL RESULTS OF HALFCHEETAH DOMAIN

We have applied the proposed adaptive reward shaping approach to the HalfCheetah Domain using different RL algorithms, including DDPG (Lillicrap et al., 2016), PPO (Schulman et al., 2017) and A2C (Mnih et al., 2016). Since none of the baselines is compatible with PPO and A2C, we only report the results of DDPG in Section 5. Here we provide additional results as shown in Figure 5. We use the same normalization factors (i.e., the maximum discounted return possible on a task) for all three algorithms.

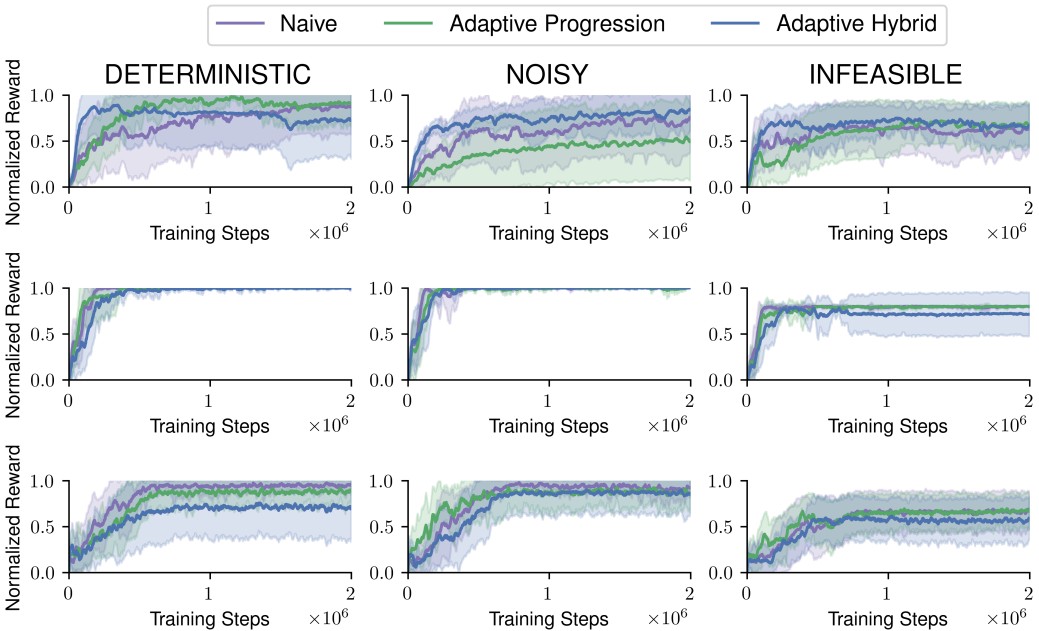

Figure 5: Results of using different RL algorithms (top row: DDPG, middle row: PPO, bottom row: A2C) in various HalfCheetah environments.

Comparing results of the three RL algorithms, we observe that DDPG exhibits relatively higher variance than others. One possible explanation for this phenomenon is rooted in the training process. DDPG is an off-policy RL algorithm, relying heavily on a replay buffer and exploration driven by control noise. In our experiments, we used a replay buffer with a capacity of $10^6$ while sampling only 100 experiences for each update. This can introduce significant randomness during the training, since the majority of samples within the large replay buffer does not yield positive rewards. Exploration also contributes to the randomness. By contrast, PPO and A2C are on-policy RL algorithms, where updates depend solely on the current policy. Consequently, these algorithms tend to maintain their behavior once the current policy achieves partial completion of the assigned task. Additionally, PPO incorporates a stabilizing technique which can help reduce the variance.

Comparing among the proposed reward functions, we find that the Naive reward function achieves comparable performance with the other two in all HalfCheetah environments. Recall from Section 5 that Naive reward function usually yields the worst performance in some other domains. One possible explanation is that the HalfCheetah task has a special structure, where each subgoal requires moving forward by the same distance. Naive reward function assigns reward 1 for completing each subgoal, which helps to keep the consistency of the learning process.

A.2 HYPERPARAMETERS OF EXPERIMENTS

For baseline methods, we used the same setup described in (Camacho et al., 2019) for QRM and (Icarte et al., 2022) for CRM.

Our implementation was built upon OpenAI Stable-Baselines3 (Raffin et al., 2021). We used their default settings of RL algorithms unless mentioned below.

**Office World.** We set the discount factor as $\gamma = 0.95$ and terminate the episode after 100 steps if the task was not completed. We set $N = 25$, $\mu = 0.5$, $\theta = 36$ and $\eta_0 = 0.01$ for the adaptive reward shaping. We used *linear-DQN without bias* for RL using hyperparameters as follows.

- (Naive, Adaptive Progression) RL agent was trained with the initial learning rate 1. Only the final learning rate was set differently: 0.5 and 1 for Naive and Adaptive Progression, respectively. In each step, the network was updated using 1 sampled experience from a replay buffer of size 1 for one time.
- (Adaptive Hybrid) RL agent was trained with the initial learning rate 1 and the final learning rate 0.5. In each step, the network was updated using 8 sampled experience from a replay buffer of size 8 for three time.

**Taxi World.** We set the discount factor as $\gamma = 0.9$ and terminate the episode after 200 steps if the task was not completed. We set $N = 100$, $\mu = 0.5$, $\theta = 26.83$ and $\eta_0 = 0.005$ for the adaptive reward shaping. We used *linear-DQN without bias* for RL using hyperparameters as follows.

- (Naive, Adaptive Progression) RL agent was trained with the same initial and final learning rate of 1. In each step, the network was updated using 2 sampled experience from a replay buffer of size 2 for three time.
- (Adaptive Hybrid) RL agent was trained with the initial learning rate 1 and the final learning rate $10^{-5}$. In each step, the network was updated using 32 sampled experience from a replay buffer of size 32 for three time.

**Water World.** We set the discount factor as $\gamma = 0.9$ and terminate the episode after 600 steps if the task was not completed. We set $N = 1000$, $\mu = 0.5$, $\theta = 36$ and $\eta_0 = 0.005$ for the adaptive reward shaping. We used *DDQN* for RL using hyperparameters as follows.

- (Naive, Adaptive Progression) RL agent was trained with the initial learning rate $10^{-5}$ and the final learning rate $10^{-6}$. We used a feed-forward network with 2 hidden layers and 256 units with ReLU activation function per layer. In each step, the network was updated using 32 sampled experience from a replay buffer of size $50,000$ for four time. We updated the target network every $1,000$ step.
- (Adaptive Hybrid) In each step, the network was updated using 1024 sampled experience. All other settings are the same as naive and adaptive progression.

**HalfCheetah.** We set the discount factor as $\gamma = 0.99$ and terminate the episode after $1,000$ steps if the task was not completed. For the adaptive reward shaping, we set $N = 1000$ for DDPG, $N = 100$ for PPO, $N = 500$ for A2C, and $\mu = 0.5$, $\theta = 15$ and $\eta_0 = 0.005$ for all. We describe hyperparameters used for each RL algorithm as follows.

- (DDPG) RL agent was trained with the initial learning rate $10^{-3}$, the final learning rate $10^{-4}$, and Polyak update coefficient 0.01. We used a feed-forward network with 2 hidden layers and 256 units with ReLU activation function per layer. We used the same setting for naive, adaptive progression, and adaptive hybrid reward functions.
- (PPO) RL agent was trained with the initial learning rate $3 \times 10^{-4}$, the final learning rate $10^{-5}$, entropy coefficient $10^{-3}$, and maximum gradient norm 0.1. We used a feed-forward network with 2 hidden layers. The first layer has 300 units, and the second layer has 400 units with ReLU activation function per layer. We used batch size 128 for Naive and Adaptive Progression, and batch size 256 for Adaptive Hybrid. Other settings are the same for all three reward functions.

- (A2C) RL agent was trained with the initial learning rate $7 \times 10^{-4}$, the final learning rate $10^{-5}$, entropy coefficient $10^{-3}$, and maximum gradient norm 0.1. We used a feed-forward network with 2 hidden layers. The first layer has 200 units, and the second layer has 300 units with ReLU activation function per layer. With Naive and Adaptive Progression reward functions, we trained the RL agent with 1024 number of steps to run for each environment per update, and used batch size 128. With Adaptive Hybrid reward function, we used 2048 number of steps per update and batch size 256. Other settings are the same for all three reward functions.

