# OpenReview forum: "Logic-Based Adaptive Reward Shaping for Reinforcement Learning"
_ICLR.cc/2024/Conference — ICLR 2024 Conference Withdrawn Submission_

### Official Review · Reviewer_XaAF · 2023-10-30

**Soundness:** 2 fair
**Presentation:** 3 good
**Contribution:** 2 fair
**Rating:** 3
**Confidence:** 4

**Summary:**

This paper studies the problem of generating shaped rewards for tasks specified using DFAs. The authors propose two methods for generating non-sparse rewards: the first method uses a distance-based progress function (where the distance to completion is defined for each DFA state based on how far the state is from the set of final states) and the second method includes a penalty for staying in the same DFA state in addition to progress function based rewards for transitioning between DFA states. Since the hardness of various transitions in the DFA is environment specific, the authors propose a way to adapt to the environment by adjusting the distance function during training. The effectiveness of the proposed reward shaping schemes is demonstrated using experiments in a variety of environments.

**Strengths:**

- Reward shaping for complex long-horizon tasks is an important and challenging problem. In particular, reward shaping mechanisms that only rely on the structure of the DFA do not take into account the difficulty of achieving various DFA transitions in the environment which often leads to reward functions that cause the agent to learn suboptimal policies which can make some progress but never achieve the goal. The proposed method for adapting the reward function during training is really interesting and provides a potential solution to this problem.
- The proposed method does not require any additional input from the user (other than the task specification) which is an improvement to some existing methods in the literature that rely on user-specified rewards for subtasks.
- The experimental results look promising and show that the adaptive reward mechanism can be used to quickly learn to perform interesting tasks.

**Weaknesses:**

- _Ad-hoc nature of the proposed technique and lack of ablations._ The adaptive reward shaping method modifies the distance (and the progress) function (e.g., it penalizes DFA states where the agent seems to get stuck at during training) and then modifies the reward function to use the maximum value of the reward based on the original and the new progress functions. This choice is not particularly well motivated beyond the demonstration of the method on the running example. It’s unclear why such an approach should work in the general case. Also, there is no ablation study validating the choices made in the definition of the adaptive reward shaping procedure, e.g.,  using the maximum, the way the progress function is updated etc.

- _The rewards are still sparse within a DFA state._ Although intermediate rewards are assigned for making progress in the specification DFA, the agent receives a constant reward for each step it remains in the same DFA state. This causes the agent to learn to transition between DFA states from sparse rewards which can require a lot of samples.

- _Lack of comparison to state-of-the-art RL from temporal specs._ There has been a lot of research on learning policies to perform tasks specified using temporal specifications including some hierarchical approaches. I believe that including empirical comparisons to more recent approaches (e.g., HRM+RS [1] and DIRL [2]) could significantly strengthen the paper.

[1] Icarte, Rodrigo Toro, et al. "Reward machines: Exploiting reward function structure in reinforcement learning." Journal of Artificial Intelligence Research 73 (2022): 173-208.
[2] Jothimurugan, Kishor, et al. "Compositional reinforcement learning from logical specifications." Advances in Neural Information Processing Systems 34 (2021): 10026-10039.

**Questions:**

1. Relating to the first point in the weaknesses section, I was wondering about a specific scenario. Suppose the DFA has a path $q_0\to q_1\to q_2\to q_3$ where $q_3$ is a final state and $q_2\to q_3$ is not physically possible in the environment. If the agent follows this path during training and gets stuck at $q_2$ often, the new distance function $d'$ after an update will be very high for $q_2$ while remaining relatively unchanged for $q_0$ and $q_1$. In this case, the new progress value for $q_1\to q_2$ will be zero and hence according to equation 8, the new reward will remain unchanged. The agent might still be incentivized to reach $q_2$. So the adaptive reward shaping is not really helping much in this scenario? In general, what are the scenarios where the adaptive reward shaping will be really helpful in encouraging the agent to satisfy the given specification?
1. In example 3, the discount factor is less than 1. Since the horizon H is finite, setting the discount factor to 1 is possible and in that case, the naïve reward will prefer $\pi_2$ over other policies. Is it true that the naïve reward function is sufficient when the discount factor is 1?
1. What is a trap state?

---

### Official Review · Reviewer_r63y · 2023-10-31

**Soundness:** 1 poor
**Presentation:** 3 good
**Contribution:** 2 fair
**Rating:** 5
**Confidence:** 4

**Summary:**

This paper addresses the problem of learning Reinforcement Learning (RL) policies that satisfy a given Linear Temporal Logic (LTL) specification. By devising an adaptive reward shaping technique, the introduced algorithm shapes the reward in a more informative manner than prior work using tools such as Reward Machines [1], which is demonstrated empirically over a range of environments.

**Strengths:**

- Presents an intuitively structured way to shape reward based on an LTL specification.
- Considers a range of different environments and applies the same reward shaping as a ‘plug-in’ to different RL algorithms including DDPG, PPO, DQN and A2C.

**Weaknesses:**

- Misses key references to related literature [2,3]. Additionally, a comparison with QRM and CRM alone could be considered insufficient; HRM was also introduced in [1] which yielded superior performance in Half-Cheetah (a similar task as considered in paper).
- Being primarily a reward shaping technique justified empirically, this work provides no formal guarantees on reward construction leading to policies satisfying the specification. It would be useful to discuss this in more a formal manner (like in [4]) while introducing guarantees on the optimal policy w.r.t. the new reward function.
- Little is said about the exact formulation of the specifications considered. A study of how the specification complexity affects policy discovery was not presented. This would bring in question the limiting cases of the algorithm.


References:

[1] Reward Machines: Exploiting Reward Function Structure in Reinforcement Learning, Icarte et al., 2022

[2] Policy Optimization with Linear Temporal Logic Constraints, Voloshin et al., 2022

[3] Eventual Discounting Temporal Logic Counterfactual Experience Replay, Voloshin et al., 2023

[4] A Composable Specification Language for Reinforcement Learning Tasks, Jothimurugan et al., 2020

**Questions:**

1. Would the authors be able to provide some experimental details including HRM[1]? If not, why is this not feasible?
2. Can the authors provide any formal guarantees on their constructed reward functions?
3. Could the authors include the exact formulation of the specifications considered (perhaps in the appendix)?
4. Do the CRM and QRM baselines use reward shaping as discussed in [1]?

---

### Official Review · Reviewer_RWQj · 2023-10-31

**Soundness:** 2 fair
**Presentation:** 2 fair
**Contribution:** 2 fair
**Rating:** 3
**Confidence:** 3

**Summary:**

The paper proposes a Reinforcement Learning framework using Linear Temporal Logic. The paper aims to address the issue of sparse reward. The two major contributions are task progression and adaptive reward shaping. , and the proposed framework is tested on OpenAi's gym environments.

**Strengths:**

The formulation of the LTL-based Deterministic Finite Automata can be used to track the progression by checking the state of each sub-task. The reward shaping ensures the distance of acceptance in Autoama is dynamically updated to ensure the agent can satisfy the LTL specification.

**Weaknesses:**

My biggest concern is the RL algorithms that are used in this paper, such as DDPG and DQN, are mode-free based learning algorithms. However, it's unclear if the construction of the  DFA-MDP requires the knowledge of the transition function, and therefore, it requires the model of the system. If my assumption is true, then the framework defeats the purpose of using model-free RL, as many other efficient methods can solve the same problems.

In addition, the presentation of the paper needs work. There are too many examples, and multiple hyperparameters need to be determined. It is unclear whether the proposed framework only fits specific scenarios and requires heavy tuning.

**Questions:**

Can the proposed DFA-MDP scale, as well as the number of sub-tasks, become large? What is the time and memory complexity of constructing it?

For the HalfCheeta example, the author claims the high variance of the DDPG causes the performance to be similar to the naive approach. Has the author tried techniques to reduce the variance, such as control regularization, to support the claim? If it's true, then the author should state under which environment/task scenario the proposed algorithm can outperform the naive approach.

---

### Official Review · Reviewer_JMof · 2023-11-01

**Soundness:** 1 poor
**Presentation:** 2 fair
**Contribution:** 1 poor
**Rating:** 3
**Confidence:** 4

**Summary:**

The paper considers the well-known yet challenging problem of reward design in reinforcement learning and proposes various methodologies from formal task specifications. The proposed approaches builds an automaton from the spec and use a distance function from the literature to define the reward. The authors present adaptive and non-adaptive update scheme for refining the reward during training. Their primary objective is to tackle reward sparsity without relying on specific algorithms. The paper’s strengths lie in addressing a pertinent issue within the learning community, providing algorithm-agnostic solutions, and proposing a novel adaptive reward update scheme.

After review, I recommend rejecting this submission due to (1) the insufficient theoretical grounding of the proposed methodology and (2) the lack of a robust empirical evaluation, which limits its practical justification.

**Strengths:**

The paper addresses a relevant problem in the field of reinforcement learning and considering frameworks from formal languages is interesting (yet not novel) for the community.

A good point of this work is aiming for solutions that are not tied to specific algorithms, targeting versatility.

**Weaknesses:**

My main criticism is about the lack of theoretical grounding and disconnection with the existing potential-based shaping literature.

Changing the reward design changes the underlying MDP and furthermore changing the reward during training (as in the adaptive case) makes the overall objective a moving target. This lacks a clear justification and leads to uncertainties regarding the entire problem setting. Moreover, the definition of the reward signals is not connected with the task satisfiability of the LTL spec, so the optimization of a policy with maximization of the cumulative reward does not lead to optimal policies, in the sense of task satisfaction. A better characterization of optimality and a formalization of the problem statement would improve the clarity of this work.

The experiments are limited in scope, consider two baselines, and do not convincingly demonstrate the superiority of the proposed approach. The experiments on environments with increasing level of noise is interesting but not fully analysed, resulting in poor clarity on why studying these different dynamics and why certain algorithms work better in diverse settings.

**Questions:**

Here, a few notes and questions that would improve the quality of this work:
- Related work, Li et al. 2017: The authors assert limitations of the Li et al. method to Markovian rewards, in contrast to the proposed method capable of generating non-Markovian rewards. However, the related work supports TLTL with until operators, defines returns over trajectories for evaluating non-Markovian rewards, and showcases applicability in specifications with non-markovian operators.
- Related work, Icarte et al. 2022: The authors suggest that the Icarte et al. method suffers from reward sparsity. However, the related work supports reward shaping to generate denser signals, demonstrated in both discrete and continuous control problems.
- Def. of progression in Eq. 2: The rationale for evaluating progression by checking the absence of cycles seems problematic for automatons with numerous cycles. For example, lets consider an automaton with states q0, q1, q2, transitions q0->q1, q1->q0, q1->q2, q2->q1, initial state q0 and accepting state q2. As per the current progression function, all transitions might be evaluated as 0. To circumvent assigning positive rewards or progression in cycles, Ng et al. (2000) proposed potential-based reward shaping.
- The term "optimal behavior" is recurrently used in the manuscript, yet a clearer characterization of what constitutes optimality would enhance comprehension. For instance, in the case of an infeasible task where neither behavior achieves the goal, the authors suggest one behavior is preferable to the other. It would be beneficial to formalize the condition for optimality or any preference relation concerning the task specification.

Minor:
- The examples 3 and 4 end up with the same calculation, despite different reward design are adopted. This makes the examples less clear.
- Why markovian rewards are only possible if |Q|=1?
- Why cannot use reward machines on other algorithms? Is there any fundamental limitations or lack of implementation?